# Sex-Based Differences in Asthma: Pathophysiology, Hormonal Influence, and Genetic Mechanisms

**DOI:** 10.3390/ijms26115288

**Published:** 2025-05-30

**Authors:** Richard Borrelli, Luisa Brussino, Luca Lo Sardo, Anna Quinternetto, Ilaria Vitali, Diego Bagnasco, Marzia Boem, Federica Corradi, Iuliana Badiu, Simone Negrini, Stefania Nicola

**Affiliations:** 1Department of Medical Sciences, University of Turin, 10128 Turin, Italy; richard.borrelli@unito.it (R.B.); luisa.brussino@unito.it (L.B.); marzia.boem@unito.it (M.B.); simone.negrini@unito.it (S.N.); 2Immunology and Allergy Unit, AO Ordine Mauriziano di Torino, 10128 Turin, Italy; llosardo@mauriziano.it (L.L.S.); anna.quinternetto@gmail.com (A.Q.); ilaria.vitali@unito.it (I.V.); fcorradi@mauriziano.it (F.C.); ibadiu@mauriziano.it (I.B.); 3UO Clinica Malattie Respiratorie e Allergologia, IRCCS-AOU San Martino, 16132 Genova, Italy; diego.bagnasco@dimi.unige.it

**Keywords:** sex, women, asthma, estrogens, hormones, severe asthma, miRNA, IL-4, IL-13, FeNO, FEV1, obesity, TLR7, TLR8, progesterone, female, male

## Abstract

Asthma is a chronic inflammatory disease characterized by airway hyperresponsiveness, variable airflow obstruction, and persistent inflammation. While its pathophysiology is well established, growing evidence highlights significant sex-based differences in its prevalence, severity, and treatment response. Epidemiological studies indicate that asthma is more common in prepubertal boys but shifts toward a female predominance after puberty, with adult women experiencing higher morbidity and greater healthcare utilization. These disparities suggest a crucial role for sex hormones, genetic predisposition, and epigenetic regulation in asthma pathogenesis. Sex hormones modulate immune responses, contributing to disease progression. Estrogen enhances type 2 inflammation, increases eosinophilic infiltration, and upregulates IL-4 and IL-13 expression, leading to greater airway hyperreactivity in women. Additionally, progesterone fluctuations correlate with perimenstrual asthma exacerbations, while testosterone appears to exert a protective effect by dampening Th2-driven inflammation and airway remodeling. These hormonal influences contribute to sex-specific asthma phenotypes and treatment responses. Genetic and epigenetic factors further shape sex-related differences in asthma. The X chromosome harbors immune-regulatory genes, including TLR7 and TLR8, which amplify inflammatory responses in females. The sex-dependent expression of IL13 and ORMDL3 influences eosinophilic inflammation and airway remodeling. Epigenetic modifications, such as DNA methylation and microRNA regulation, further impact immune activation and corticosteroid responsiveness. For instance, Let-7 miRNAs modulate IL-13 expression, contributing to sex-specific inflammatory profiles. Environmental factors, including air pollution, obesity, and diet, interact with hormonal and genetic influences, exacerbating sex disparities in asthma severity. Obesity-related metabolic dysfunction promotes systemic inflammation, airway remodeling, and steroid resistance, disproportionately affecting women. Given these complex interactions, sex-specific approaches to asthma management are essential. Personalized treatment strategies targeting hormonal pathways, immune regulation, and metabolic health may improve outcomes for both men and women with asthma. Future research should focus on sex-based therapeutic interventions to optimize disease control and mitigate healthcare disparities.

## 1. Introduction

Asthma is a chronic inflammatory condition marked by airway hyperresponsiveness, variable airflow obstruction, and persistent inflammation. It manifests through symptoms such as wheezing, shortness of breath, chest tightness, and coughing, which can fluctuate in severity and frequency among individuals [1]. According to the Global Initiative for Asthma (GINA), it is defined as a heterogeneous disease primarily characterized by chronic airway inflammation and a history of respiratory symptoms that change over time in both intensity and frequency, often accompanied by variable expiratory airflow limitation [2]. Despite significant advancements in the understanding of its mechanisms, asthma remains a major public health concern, affecting millions worldwide [3].

Sex and gender are often used interchangeably, yet they refer to distinct concepts. Sex is determined by biological factors, such as reproductive anatomy and genetic makeup, whereas gender encompasses the social and cultural influences that shape an individual’s identity and roles within society. However, these two aspects are not entirely separate, as both contribute to variations in health, disease susceptibility, and treatment responses [4]. One crucial aspect that has garnered increasing attention in recent years is the influence of sex on asthma prevalence, severity, and response to treatment.

This review explores sex-based differences in asthma by analyzing the role of hormones, genetic predispositions, immune responses, and their cumulative impact on disease presentation and progression.

## 2. Epidemiology

Asthma remains a globally prevalent and clinically significant chronic respiratory disease, affecting approximately 300 million individuals worldwide [5]. Its epidemiology is highly variable across different geographical regions, with prevalence rates influenced by genetic, environmental, and socio-economic factors. In high-income countries, asthma affects around 10% of the adult population and up to 20% of children, whereas in low- and middle-income countries (LMICs), prevalence is often underestimated due to limited access to healthcare services and diagnostic tools [6]. The burden of asthma extends beyond its direct impact on patients, placing considerable strain on healthcare systems and contributing to substantial economic costs associated with hospitalizations, emergency visits, and loss of productivity [3,7].

Among individuals with asthma, a subset presents with what is classified as difficult-to-treat asthma, a condition in which symptoms remain poorly controlled despite adherence to medium-to-high doses of inhaled corticosteroids (ICS) in combination with long-acting beta-2 agonists (LABAs) or systemic corticosteroids [1,2,8]. In addition to adherence and inhaler technique, the presence of comorbid conditions often complicates the management of difficult-to-treat asthma. Chronic rhinosinusitis, gastroesophageal reflux disease (GERD), obstructive sleep apnea (OSA), and vocal cord dysfunction (VCD) are among the most common comorbidities that can mimic or exacerbate asthma symptoms [9].

A smaller subset of patients with asthma is classified as having severe asthma, a condition that remains uncontrolled despite adherence to high-dose ICS-LABA therapy and the optimal management of contributing factors. Severe asthma represents a distinct clinical entity with a significant disease burden, characterized by persistent symptoms, frequent exacerbations, and an increased risk of life-threatening complications [7,8]. Unlike difficult-to-treat asthma, where modifiable factors often contribute to poor disease control, severe asthma persists despite optimal therapy and requires advanced interventions for adequate management.

Asthma disproportionately affects certain populations, with significant disparities observed in disease severity, healthcare access, and treatment outcomes. Pediatric asthma remains a major global health concern, with a substantial proportion of cases developing in early childhood, often triggered by viral respiratory infections, environmental allergens, or early-life exposure to tobacco smoke [5].

In recent decades, the incidence of asthma has risen in many industrialized nations, a trend often attributed to environmental and lifestyle changes. The “hygiene hypothesis” suggests that reduced early-life exposure to microbes alters immune system development, leading to an increased susceptibility to allergic diseases, including asthma [10]. Additionally, rapid urbanization has been associated with an increased prevalence of asthma, largely due to higher exposure to environmental pollutants such as nitrogen dioxide, particulate matter, and indoor allergens; moreover, the role of air pollution in worsening asthma control is well established, with evidence linking exposure to fine particulate matter and ozone to increased airway inflammation, exacerbations, and long-term lung function decline [11,12].

The role of diet and metabolic health has also emerged as a potential contributor to asthma pathophysiology; obesity has long been recognized as a risk factor for asthma, but emerging research suggests that metabolic dysfunction, rather than adiposity alone, plays a crucial role in shaping disease severity, response to treatment, and long-term outcomes [13]. The interplay between diet, systemic inflammation, immune dysregulation, and airway remodeling creates a complex biological framework that contributes to asthma exacerbations and diminished lung function [14].

Asthma exhibits distinct patterns of incidence, prevalence, and severity across the lifespan, with notable differences between males and females. While asthma is commonly considered a childhood-onset disease, it can develop at any stage of life. However, the distribution of asthma between sexes shifts considerably over time, suggesting a complex interplay of biological, genetic, environmental, and socio-economic factors in disease pathogenesis.

In early childhood, asthma is significantly more prevalent in boys than in girls, with boys under the age of 13 experiencing higher incidence, prevalence, and hospitalization rates. Epidemiological data indicate that asthma prevalence in boys is 11.9%, compared to 7.5% in girls [15]. Additionally, boys are twice as likely as girls to be hospitalized for an asthma exacerbation [16,17]. This disparity is evident in both developed and developing countries and is thought to be driven by multiple factors, including anatomical and immunological differences. Male children tend to have narrower airways relative to lung volume, which may contribute to increased airway resistance and a higher likelihood of experiencing asthma symptoms and exacerbations [18]. Furthermore, as elaborated in the subsequent sections, immune system development differs between the sexes, with boys exhibiting a more pronounced Type 2 inflammatory response in early life, predisposing them to a higher burden of allergic diseases, including asthma.

Despite the higher prevalence of asthma in prepubescent males, their symptoms often improve with age, leading to a significant decline in asthma incidence during adolescence [5]. This remission is not universally observed but is sufficiently common to contribute to the shifting epidemiological pattern seen in later life stages. As individuals transition into adolescence, the male predominance in asthma burden diminishes and is eventually reversed; in fact, by puberty, females begin to experience higher rates of asthma incidence and persistence compared to their male counterparts. Early menarche (before 11 years of age) has been associated with an increased risk of developing asthma [19]. This trend continues into adulthood, where women consistently demonstrate a greater prevalence of asthma than men. By adulthood, asthma prevalence in women is 9.6%, compared to 6.3% in men [15]. Furthermore, women are three times more likely than men to be hospitalized for an asthma-related event [5,20].

The sex disparity in asthma burden narrows again around the fifth decade of life, with some studies suggesting a secondary increase in asthma incidence among men during middle age [4,15,21]. This trend may be influenced by occupational exposures, smoking history, and age-related changes in immune function and airway remodeling. Additionally, menopause represents a significant physiological transition that may alter asthma prevalence and severity in women. Some studies suggest that asthma prevalence decreases in postmenopausal women compared to premenopausal women [22]. However, variable findings exist, with some research indicating increased asthma symptoms during the menopausal transition [23,24].

While asthma-related healthcare utilization is higher in children overall, morbidity and mortality due to asthma are disproportionately higher in adults, particularly among women [5,25]. Adult females are more likely than their male counterparts to experience severe asthma, characterized by frequent exacerbations, reduced lung function, and increased dependence on healthcare resources [25]. In developed countries, this pattern is reflected in higher healthcare utilization rates among women aged 23 to 64, mirroring the peak in asthma prevalence observed during these years [26]. Additionally, approximately 30–40% of women with asthma report premenstrual worsening of asthma symptoms, which is associated with increased inflammation and a higher frequency of emergency department visits [27].

The intersection of hormonal changes, genetic predisposition, and environmental exposures shapes the epidemiological landscape of asthma, reinforcing the need for sex-specific approaches in asthma management and treatment.

## 3. Pathophysiology

The pathophysiology of asthma is primarily driven by dysregulated immune responses, which can be broadly categorized into type 2 (T2-high) and non-type 2 (T2-low) inflammation, each associated with distinct cellular and molecular mechanisms [28].

T2-high asthma is characterized by an exaggerated type 2 immune response that drives airway inflammation, remodeling, and hyperresponsiveness. This inflammatory cascade is mediated by Th2 cells and group 2 innate lymphoid cells (ILC2s), both of which secrete interleukins that amplify the immune response [29]. IL-4 plays a central role by promoting B-cell differentiation and class switching to IgE production, a crucial step in allergic sensitization [30]. The binding of allergen-specific IgE to mast cells leads to their degranulation upon allergen exposure, releasing histamine, prostaglandins, and leukotrienes that contribute to bronchoconstriction, vascular permeability, and mucus secretion [31]. IL-5 is instrumental in recruiting and activating eosinophils, which infiltrate the airway tissue, release cytotoxic granules, and further amplify tissue damage and inflammation [32]. IL-13, another key cytokine in T2-high asthma, enhances goblet cell hyperplasia and mucus hypersecretion while promoting airway smooth muscle contraction and fibrosis, which contribute to airway remodeling [33]. The eosinophilic inflammation observed in this endotype is often associated with increased airway hyperresponsiveness, leading to recurrent symptoms and exacerbations. Patients with T2-high asthma typically respond well to inhaled corticosteroids, which suppress the type 2 inflammatory cascade, and to biologic therapies targeting IL-4, IL-5, and IL-13 [34,35,36,37].

In contrast, T2-low asthma is characterized by neutrophilic or paucigranulocytic inflammation and is often less responsive to corticosteroid therapy [38]. The mechanisms underlying this phenotype are less well defined but appear to involve a shift towards Th1- and Th17-mediated inflammation. Th1 cells produce interferon-gamma (IFN-γ), a cytokine that enhances macrophage activation and promotes a pro-inflammatory response that is commonly observed in non-allergic, infection-driven, or pollutant-exacerbated asthma. Th17 cells, on the other hand, secrete IL-17, which plays a role in neutrophil recruitment and activation, further driving airway inflammation and structural damage [38]. Neutrophilic inflammation in T2-low asthma is associated with increased oxidative stress, epithelial injury, and mucus [39].

Unlike eosinophilic asthma, which is often steroid-responsive, neutrophilic asthma is more resistant to corticosteroid treatment, potentially due to the involvement of alternative inflammatory pathways that do not rely on IL-4, IL-5, or IL-13 signaling.

The distinction between T2-high and T2-low asthma has significant implications for treatment and prognosis. While T2-high asthma is generally more amenable to corticosteroid therapy and biologic interventions, T2-low asthma poses a greater challenge due to its steroid resistance and poorly defined inflammatory drivers [38,40].

The chronic inflammation observed in asthma leads to progressive structural changes in the airways, collectively referred to as airway remodeling. This process is driven by persistent immune activation, tissue injury, and aberrant repair mechanisms, which ultimately result in irreversible alterations in airway architecture. Key features of airway remodeling include subepithelial fibrosis, airway smooth muscle hypertrophy and hyperplasia, goblet cell metaplasia, increased angiogenesis, and the thickening of the airway walls. These changes contribute to airflow limitation, decreased lung function, and a diminished response to bronchodilators over time [41,42].

One of the earliest manifestations of airway remodeling is subepithelial fibrosis, which results from the excessive deposition of extracellular matrix (ECM) components such as collagen and fibronectin in the subepithelial basement membrane [43]. This process is largely driven by IL-13 and transforming growth factor-beta (TGF-β), which activate fibroblasts and myofibroblasts, promoting ECM production and deposition [44]. The thickening of the reticular basement membrane, a hallmark of chronic asthma, reduces airway elasticity and contributes to fixed airflow obstruction.

Airway smooth muscle hypertrophy and hyperplasia further exacerbate airway narrowing and bronchial hyperresponsiveness. Chronic exposure to inflammatory cytokines, including IL-13 and platelet-derived growth factor (PDGF), stimulates smooth muscle proliferation and increases contractile protein expression, enhancing airway constriction in response to various stimuli [45]. The increased mass of airway smooth muscle not only contributes to bronchoconstriction but also influences the inflammatory milieu by secreting cytokines and chemokines that perpetuate immune activation.

Goblet cell metaplasia and mucus hypersecretion represent another critical aspect of airway remodeling in asthma. IL-13 plays a central role in driving goblet cell differentiation and increasing mucin gene expression, particularly MUC5AC, which leads to excessive mucus production [46]. The resulting mucus plugs contribute to airway obstruction and airflow limitation, particularly in severe asthma. In addition, impaired mucociliary clearance due to ciliary dysfunction exacerbates mucus accumulation, further increasing the risk of airway occlusion and exacerbations [47].

A fundamental consequence of these structural changes is the progressive decline in lung function observed in some patients with asthma. Unlike the reversible airflow limitation seen in early or well-controlled disease, airway remodeling contributes to fixed obstruction, limiting the effectiveness of bronchodilators and corticosteroids. This decline is more pronounced in patients with severe or poorly controlled asthma, underscoring the need for early and aggressive intervention to prevent long-term damage [48].

Airway hyperresponsiveness (AHR), a key feature of asthma, results from both the inflammatory and structural changes in the airways. AHR is defined as an exaggerated bronchoconstrictive response to various stimuli, including allergens, cold air, pollutants, and non-specific irritants [49]. The mechanisms underlying AHR involve increased sensitivity of airway smooth muscle to contractile agonists, alterations in the neural control of airway tone, and heightened immune activation. In T2-high asthma, eosinophilic inflammation and IL-13-mediated smooth muscle changes contribute to AHR [50], whereas in T2-low asthma, factors such as neutrophil-mediated inflammation, oxidative stress, and epithelial dysfunction play a more significant role [38,40].

## 4. Hormones and Asthma

Sex differences in asthma prevalence and severity change throughout life. In childhood, asthma is more prevalent in boys than girls, with a higher rate of hospitalization and exacerbations in male children [5,17]. However, during puberty, the prevalence shifts, with adult women exhibiting higher asthma rates, greater severity, and a more persistent disease course compared to men [20,21,51]. This transition suggests the significant impact of sex hormones on asthma pathophysiology. Hormones play a pivotal role in mediating the sex-based differences observed in asthma. Estrogen, progesterone, and testosterone regulate immune responses, airway inflammation, and bronchial hyperresponsiveness, influencing asthma onset and severity at different life stages.

Indeed, estrogen has been shown to enhance Th2-mediated airway inflammation, increasing eosinophilic infiltration, mucus production, and airway hyperreactivity [52,53]. Clinical and animal studies indicate that estrogen amplifies type 2 inflammation by upregulating IL-4 and IL-13, which drive airway remodeling and mucus secretion [52,54]. The increased prevalence of asthma in postpubertal females and the exacerbation of symptoms during menstruation, pregnancy, and hormone replacement therapy (HRT) further emphasize estrogen’s role [55]. Studies have reported a higher expression of CRTh2, a chemoattractant receptor expressed on Th2 cells, in women with severe asthma. The dual activation of estrogen receptor alpha (ERα) and glucocorticoid receptor (GR) has been shown to enhance CRTh2-mediated inflammation, suggesting that estrogen modulates corticosteroid response in asthma patients [56]. Estrogen can modulate airway inflammation through multiple pathways, including its effects on innate lymphoid cells (ILCs). Recent studies have demonstrated that ILC2s, which play a crucial role in allergic inflammation, are more active in females than males, contributing to increased asthma severity in women [57]. Additionally, estrogen enhances IL-17A-mediated airway inflammation, which is more pronounced in females with severe asthma compared to males [58]. Increased IL-17A expression correlates with neutrophilic inflammation, a feature of corticosteroid-resistant asthma, suggesting that estrogen’s role extends beyond type 2 inflammation [59].

On the other hand, progesterone’s impact on asthma is more complex, as it can have both pro-inflammatory and anti-inflammatory effects; while some studies suggest that progesterone reduces inflammation by modulating macrophage activity, others indicate that it may contribute to airway hyperresponsiveness by affecting smooth muscle contraction and mucus secretion [52,57]. The fluctuations in progesterone levels during the menstrual cycle correlate with worsening asthma symptoms in some women, a phenomenon termed perimenstrual asthma (PMA) [27]. Women with PMA have increased airway inflammation and require higher doses of corticosteroids for symptom control [27,60].

In contrast to estrogen and progesterone, testosterone appears to exert a protective effect against asthma, as numerous studies showed it can reduce Th2-driven inflammation, inhibit eosinophilic activity, and modulate airway remodeling [61]. Moreover, testosterone is capable of suppressing IL-13 production, decreasing mucus secretion and airway obstruction [62]. This protective role may explain why asthma prevalence declines in males after puberty and remains lower in men compared to women throughout adulthood. Castration studies in animal models have demonstrated increased airway inflammation in males, which can be reversed with testosterone replacement, further confirming its regulatory role [63].

Obesity has also been identified as a significant modifier of asthma severity, with evidence suggesting that excess adiposity contributes to systemic inflammation, increased airway resistance, and reduced responsiveness to corticosteroid therapy [14]; moreover, recent evidence has highlighted the significant influence of diet and metabolic health on asthma pathophysiology [64]. Obesity has long been recognized as a risk factor for asthma, but emerging research suggests that metabolic dysfunction, rather than adiposity alone, plays a crucial role in shaping disease severity, response to treatment, and long-term outcomes. The interplay between diet, systemic inflammation, immune dysregulation, and airway remodeling creates a complex biological framework that contributes to asthma exacerbations and diminished lung function. One of the primary mechanisms linking obesity and asthma involves low-grade systemic inflammation driven by adipose tissue. Adipose depots, particularly visceral fat, function as an active endocrine organ, secreting pro-inflammatory cytokines such as interleukin-6 (IL-6), tumor necrosis factor-alpha (TNF-α), and leptin, while reducing levels of the anti-inflammatory adipokine adiponectin [65]. Increased IL-6 levels have been correlated with asthma severity, independent of body mass index, suggesting that metabolic dysfunction itself is a key driver of inflammation in obese asthmatics [66]. Furthermore, obesity skews immune responses toward a more pronounced Th1 and Th17 phenotype, which contrasts with the traditional Th2-driven eosinophilic inflammation seen in allergic asthma. This shift may explain why many obese asthmatics exhibit a neutrophilic pattern of airway inflammation, which is often associated with steroid resistance and a more severe disease course [67]; in fact, metabolic dysregulation extends beyond inflammation and affects lung function through multiple pathways. Hyperinsulinemia, insulin resistance, and dyslipidemia contribute to airway remodeling by promoting epithelial dysfunction, smooth muscle hypertrophy, and increased oxidative stress. Insulin resistance has been implicated in airway hyperresponsiveness through its effects on airway smooth muscle contractility and extracellular matrix deposition. Moreover, hyperglycemia has been shown to enhance airway inflammation, possibly by increasing oxidative stress and impairing nitric oxide bioavailability [68].

## 5. Sex-Related Differences in Terms of Genetic and Epigenetic Factors

Beyond hormonal influences, genetic, epigenetic, and posttranscriptional regulatory mechanisms contribute to the distinct sex differences observed in asthma prevalence, severity, and immune response. A growing body of research has highlighted the role of sex-specific gene expression, differential DNA methylation patterns, histone modifications, and microRNA (miRNA) regulation in shaping asthma phenotypes in males and females. These molecular mechanisms interact with hormonal signals and environmental exposures, influencing airway inflammation, immune activation, and response to treatment.

Genetic factors play a crucial role in sex-based differences in asthma, particularly through the influence of sex chromosomes and autosomal genes that regulate immune responses and airway remodeling. The X chromosome harbors numerous immune-related genes that may contribute to the increased asthma prevalence and severity observed in females. Due to X chromosome inactivation, females maintain a unique immune profile, as certain genes escape inactivation and are expressed at higher levels compared to males [69]. Among these, *TLR7* and *TLR8* encode toll-like receptors involved in innate immune activation and are more highly expressed in females, potentially amplifying inflammatory responses in the airways [70].

In addition to X-linked genes, autosomal genes implicated in asthma susceptibility exhibit sex-dependent expression patterns. IL13, a key cytokine in type 2 inflammation, is differentially regulated in males and females, with increased expression in females contributing to stronger eosinophilic inflammation and mucus hypersecretion [71]. Similarly, *ORMDL3*, a gene involved in sphingolipid metabolism, airway hyperresponsiveness, and *ADAM33*, which plays a role in airway remodeling, demonstrates sex-dependent variation in expression, likely influenced by hormonal modulation [72,73].

Epigenetic modifications such as DNA methylation, histone modifications, and miRNA regulation add another layer of complexity to sex-based differences in asthma. DNA methylation plays a crucial role in gene expression regulation, and studies have identified distinct methylation patterns in asthma-related genes between males and females. These modifications can influence immune cell function, airway inflammation, and steroid responsiveness.

miRNAs have emerged as critical regulators of posttranscriptional gene expression, influencing cytokine production, immune cell activation, and airway remodeling. Several miRNAs have been implicated in asthma pathogenesis, with recent studies suggesting their role in mediating sex-based differences in disease susceptibility and severity.

One notable miRNA family involved in asthma regulation is the Let-7 family. Let-7 miRNAs have been found to be downregulated in asthma, and emerging evidence suggests they may contribute to sex disparities in the disease; in fact, Let-7 miRNAs regulate IL-13 expression, which, as mentioned in the pathophysiology, covers a central role in type 2 inflammation [74]. Studies have shown that *Let-7f*, *Let-7g*, and *miR-98* expression is higher in the bronchial tissue of males compared to females, potentially leading to lower IL-13 levels in males and a reduced inflammatory response [75]. In contrast, females exhibit higher levels of IL-13, thymic stromal lymphopoietin (TSLP), and soluble ST2 (sST2), indicating a more pronounced type 2 inflammatory environment [76].

Another key miRNA involved in sex-specific asthma pathophysiology is *miR-712-5p*. This miRNA has been shown to regulate inflammatory cytokine expression, a hallmark of asthma. Notably, asthma prevalence is higher in adult females, and they exhibit greater susceptibility to environmental pollutants, leading to more severe disease phenotypes. Experimental models have demonstrated that *miR-712* expression differs between males and females in response to allergen sensitization and pollutant exposure [77].

The interplay between genetic predisposition, epigenetic modifications, and posttranscriptional regulation by miRNAs highlights the complexity of sex-based differences in asthma. These molecular differences not only influence disease susceptibility but also impact treatment response, particularly in the context of corticosteroid resistance. Given that females tend to exhibit more severe asthma and a greater likelihood of steroid-resistant disease, future therapeutic strategies should consider targeting sex-specific pathways.

Biologic therapies that modulate IL-13, TSLP, and ILC2 activation may be particularly beneficial in females with severe asthma, whereas targeting ST2L signaling and modifying Let-7 or *miR-712* expression may provide novel treatment avenues for males. Understanding the mechanisms by which miRNAs and epigenetic modifications influence immune activation in asthma could pave the way for personalized medicine approaches that take sex differences into account.

## 6. Clinical Implications and Treatment Considerations

The study of sex-based differences in asthma reveals a complex interplay of hormonal, genetic, and immune mechanisms that contribute to variations in disease prevalence, severity, and treatment response, as shown in Table 1. Epidemiological data indicate that asthma is more common in boys during childhood but becomes more prevalent and severe in women after puberty, highlighting the influence of sex hormones on immune function and airway inflammation. Estrogen has been shown to enhance type 2 inflammation and contribute to increased asthma severity in females, whereas testosterone appears to exert a protective effect.

Genetic and epigenetic factors further shape these sex differences, with variations in immune-related gene expression, DNA methylation, and microRNA regulation contributing to differential immune responses and airway remodeling in males and females. These molecular mechanisms not only influence asthma susceptibility but also impact treatment outcomes, particularly regarding corticosteroid responsiveness, which tends to be lower in women with severe asthma.

A greater understanding of these sex-specific differences is crucial for developing personalized treatment approaches. Current therapeutic strategies largely follow a one-size-fits-all model, yet growing evidence suggests that sex should be considered a fundamental variable in asthma management. Future research should focus on tailoring interventions to address the unique immunological and hormonal factors that influence asthma pathophysiology in men and women. By integrating sex-specific considerations into clinical practice, it may be possible to optimize treatment efficacy and improve long-term outcomes for patients with asthma.

## Figures and Tables

**Table 1 ijms-26-05288-t001:** Different features of asthma according to gender and age.

Aspect	Males	Females
**Childhood Prevalence**	Higher	Lower
**Adulthood Prevalence**	Lower	Higher
**Hormonal Influence**	Protective effect of testosterone	Pro-inflammatory effects of estrogen and progesterone
**Immune Response**	Reduced Th2 activation post puberty	Increased Th2 response via IL-13, ILC2, and IL-17A activity
**Genetic Factors**	Higher expression of Let-7 family miRNAs	Enhanced expression of TLR7, TLR8, IL-13
**Epigenetics/miRNA**	Suppressed IL-13 via miR-98, Let-7	Higher IL-13, TSLP, ST2 due to miRNA differences
**Corticosteroid Response**	Better responsiveness	Reduced efficacy, especially in severe asthma
**Airway Remodeling**	Less prominent	More pronounced due to hormonal and metabolic interactions

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
