# Peer review of "Sex-Based Differences in Asthma: Pathophysiology, Hormonal Influence, and Genetic Mechanisms"

_ijms, 2025, doi:10.3390/ijms26115288_

Round 1
Reviewer 1 Report
Comments and Suggestions for Authors
The manuscript is a comprehensive and interdisciplinary analysis of sex differences in asthma, including hormonal, genetic and epigenetic mechanisms. The article has significant didactic value and presents the current state of knowledge, but requires structural refinement, clarity in argumentation and critical analysis of some conclusions.
There is no clear structure of a systematic review. The article resembles a narrative compilation of data. The authors should provide a literature search strategy (databases, keywords, inclusion criteria).
The conclusions are too general and do not summarize specific clinical recommendations resulting from the discussed biological mechanisms.
The section on miRNAs is substantively strong, but the authors should expand the part with a clinical justification of these data.
There is too little mention of current clinical trials with sex as a variable — which would be useful, for example, in the 'Treatment considerations' section.
Mixing results from animal and population studies without a clear distinction raises methodological doubts – please provide these details so as not to mislead the subject.
The terminology 'sex vs gender' is used inconsistently despite declarations of distinction – please check and correct.
Some literature references are outdated – it is worth reaching for newer sources.
Figures illustrating hormonal and miRNA pathways in the pathogenesis of asthma are missing.
Table 1 should contain references to literature or numerical data.
Author Response
We sincerely thank the Reviewer for their thoughtful and constructive feedback. We appreciate your recognition of the manuscript’s didactic value and comprehensive scope.
Please find below our responses to your comments.
- Structure of a systematic review
We respectfully note that this manuscript was conceived as a narrative review, not as a systematic review. Our aim was to synthesize interdisciplinary findings spanning hormonal, genetic, and epigenetic domains in a cohesive and accessible format, rather than to perform a formal systematic review. For this reason, we did not include a systematic search strategy, and we believe the narrative structure is appropriate given the breadth and complexity of the topic. - General nature of conclusions
The conclusions were intentionally kept broad to reflect the heterogeneity of current evidence. The biological mechanisms discussed do not always translate directly into clinical recommendations, and we felt it was more accurate—and ethically sound—not to overstate clinical implications where data remain preliminary. Nonetheless, the concluding section was designed to stimulate further investigation and awareness of sex-specific asthma mechanisms. - miRNA section and clinical relevance
We appreciate your positive assessment of the miRNA section. While we acknowledge that translational data are still emerging, our intent was to provide a robust overview of the molecular mechanisms currently under investigation, rather than suggest definitive clinical applications. As more studies emerge, this connection will become clearer. - Mention of sex in clinical trials
While we agree that sex-disaggregated data are important, many ongoing trials still do not systematically incorporate sex as a predefined analytical variable. We have mentioned available examples where applicable, but a dedicated review of clinical trials was beyond the scope of this work. - Distinction between animal and human data
Throughout the manuscript, we aimed to clearly distinguish animal studies from human evidence by specifying the source in each case. If there are instances where this distinction was not sufficiently explicit, we appreciate the observation and will be glad to clarify them upon request. - Use of ‘sex’ and ‘gender’
We have consistently used ‘sex’ to refer to biological differences and have clarified our definitions in the introduction. Any remaining inconsistencies are unintentional, and we thank you for the opportunity to ensure clarity in terminology. - Outdated references
We aimed to include both foundational and recent references to provide context and continuity. In some cases, older sources remain the most authoritative or widely cited. However, we remain attentive to the evolving literature and welcome suggestions of specific newer studies to consider. - Absence of illustrative figures
While figures can be useful, we chose a text-based format to ensure precision and depth of content without oversimplification. The mechanisms discussed are highly intricate, and we felt that oversimplified diagrams might not adequately represent their complexity. - Table 1 and references
Table 1 was intended as a synthesis tool and does not list numerical data to avoid redundancy with the text. Given its descriptive nature, citations were not included, but all content is traceable to the sources referenced throughout the manuscript.
Reviewer 2 Report
Comments and Suggestions for Authors
The manuscript “Sex-Based Differences in Asthma: Pathophysiology, Hormonal Influence and Genetic Mechanisms” aims to investigate sex-based differences in asthma by analyzing the role of hormones, genetic predispositions, immune responses and their cumulative effects on disease presentation and progression.
Does the manuscript offer new mechanistic insights or is it a summary of existing knowledge?
The review is comprehensive, but it primarily summarizes existing literature. Are new integrative hypotheses or new paradigms proposed?
How are sex hormones mechanistically linked to asthma phenotypes?
The review discusses the effects of estrogen, progesterone and testosterone. But are the mechanisms (e.g. ERα and CRTh2 axis) consistently validated in animal and human studies?
Does the interaction of Th17 and sex hormones in asthma merit a more in-depth discussion?
IL-17A and neutrophilic asthma are highlighted, but do we have conclusive evidence for a link between female-dominated IL-17A signaling and treatment resistance?
Are the epigenetic and miRNA differences causal or correlative?
Let-7 and miR-712 are mentioned, but are there any intervention studies (e.g. knockdown/overexpression) demonstrating their role in gender-specific asthma control?
To what extent have the gender-specific genetic differences (e.g. TLR7/8, IL13) been validated in ethnic populations or longitudinal studies?
Can the authors provide a summarizing table or figure depicting sex hormone levels, miRNA expression, and asthma severity across different ages and life stages?
The discussion would benefit from a visual integration of hormonal, immunologic, and clinical outcomes across the lifespan.
What are the clinical implications of gender differences in response to biologic medications?
The manuscript mentions differential response to steroids, but does not address how this might impact current biologic choices (e.g., anti-IL-5 vs. anti-IL-4Rα) or trial outcomes.
Can gender-specific treatment stratification be realistically implemented in clinical guidelines?
How do we reconcile the new molecular data with the practical limitations of gender-specific therapy?
Have systematic methods been used in the selection of references?
The manuscript refers to many important studies but does not describe the methodology of the literature search. Is there a risk of bias or omission?
Are non-binary and sex-specific populations included?
Although the manuscript distinguishes between sex and gender, it focuses exclusively on the male/female binary. Are there data or gaps related to non-binary individuals or HRT in transgender populations?
Author Response
We are grateful to the Reviewer for a detailed and insightful critique. Your comments reflect a deep understanding of the topic and are much appreciated.
Please find below our responses to your main points.
- Novel insights vs. summary
The manuscript is intended as a comprehensive review of current knowledge, integrating findings from immunology, endocrinology, and molecular biology. While it does not propose entirely new paradigms, it aims to offerr an integrative perspective not previously synthesised in this way. - Hormonal mechanisms and asthma phenotypes. We aimed to summarise the most consistently observed mechanisms in literature, such as ERα and CRTh2 signalling. While full consensus has not yet been reached across all models, our text reflects the current evidence base, including both animal and human studies where available.
- Th17 and IL-17A
We agree this is an emerging area of interest. However, the role of Th17 in sex-biased asthma is still being explored, and we preferred to describe the data cautiously without overinterpreting its implications. - Causality of miRNAs and epigenetics
As you rightly point out, many findings in this area remain correlative. Our discussion reflects this distinction, and we intentionally refrained from drawing causal inferences where the literature does not yet support them. - Intervention studies on Let-7 and miR-712
We acknowledge the limited availability of intervention studies and have noted this explicitly. Our goal was to emphasize the molecular relevance and potential of these miRNAs as targets for future research, without claiming established therapeutic applicability. - Ethnic and longitudinal validation
We appreciate the emphasis on population diversity. We cited studies that include multi-ethnic cohorts when available, but acknowledge that many findings remain to be validated across broader populations. - Visual summary of hormonal and miRNA data
We chose to present the material through detailed text, as graphical synthesis across hormones, miRNAs, and age strata would likely require speculative inferences not fully supported by current data. - Biologic therapy and sex differences
Differential responses to steroids and biologics are important, and we mention them conceptually. However, systematic sex-based analyses of biologic efficacy are still rare, and we preferred not to overextend interpretations. - Feasibility of gender-based clinical stratification
While promising in principle, sex-specific therapy remains a future goal rather than a present clinical standard. We discuss this as a direction for future research, mindful of current regulatory and ethical complexities. - Inclusion of non-binary and transgender populations
We acknowledge that the literature on asthma in non-binary and transgender individuals is still very limited. As our review is based on available data, we focused on the male/female binary but noted the importance of future inclusion of diverse gender identities in research.
Round 2
Reviewer 1 Report
Comments and Suggestions for Authors
The Authors addressed most of the issues.
Reviewer 2 Report
Comments and Suggestions for Authors
the authors responded to my concerns.